# Graphical Models for Recovering Probabilistic and Causal Queries from Missing Data

**Karthika Mohan and Judea Pearl**
Cognitive Systems Laboratory
Computer Science Department
University of California, Los Angeles, CA 90024
{karthika,judea}@cs.ucla.edu

## Abstract

We address the problem of deciding whether a causal or probabilistic query is estimable from data corrupted by missing entries, given a model of missingness process. We extend the results of Mohan et al. [2013] by presenting more general conditions for recovering probabilistic queries of the form $P(y|x)$ and $P(y,x)$ as well as causal queries of the form $P(y|do(x))$. We show that causal queries may be recoverable even when the factors in their identifying estimands are not recoverable. Specifically, we derive graphical conditions for recovering causal effects of the form $P(y|do(x))$ when $Y$ and its missingness mechanism are not d-separable. Finally, we apply our results to problems of attrition and characterize the recovery of causal effects from data corrupted by attrition.

## 1   Introduction

All branches of experimental science are plagued by missing data. Improper handling of missing data can bias outcomes and potentially distort the conclusions drawn from a study. Therefore, accurate diagnosis of the causes of missingness is crucial for the success of any research. We employ a formal representation called 'Missingness Graphs' (m-graphs, for short) to explicitly portray the missingness process as well as the dependencies among variables in the available dataset (Mohan et al. [2013]). Apart from determining whether recoverability is feasible namely, whether there exists any theoretical impediment to estimability of queries of interest, m-graphs can also provide a means for communication and refinement of assumptions about the missingness process. Furthermore, m-graphs permit us to detect violations in modeling assumptions even when the dataset is contaminated with missing entries (Mohan and Pearl [2014]).

In this paper, we extend the results of Mohan et al. [2013] by presenting general conditions under which probabilistic queries such as joint and conditional distributions can be recovered. We show that causal queries of the type $P(y|do(x))$ can be recovered even when the associated probabilistic relations such as $P(y,x)$ and $P(y|x)$ are not recoverable. In particular, causal effects may be recoverable even when $Y$ is not separable from its missingness mechanism. Finally, we apply our results to recover causal effects when the available dataset is tainted by attrition.

This paper is organized as follows. Section 2 provides an overview of missingness graphs and reviews the notion of recoverability i.e. obtaining consistent estimates of a query, given a dataset and an m-graph. Section 3 refines the sequential factorization theorem presented in Mohan et al. [2013] and extends its applicability to a wider range of problems in which missingness mechanisms may influence each other. In section 4, we present general

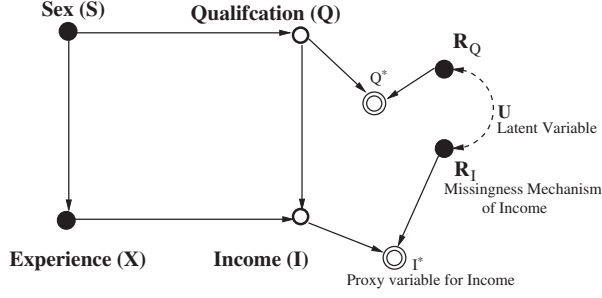

Figure 1: Typical m-graph where $V_o = \{S, X\}$, $V_m = \{I, Q\}$, $V^* = \{I^*, Q^*\}$, $R = \{R_i, R_q\}$ and $U$ is the latent common cause. Members of $V_o$ and $V_m$ are represented by full and hollow circles respectively. The associated missingness process and assumptions are elaborated in appendix 10.1.

algorithms to recover joint distributions from the class of problems for which sequential factorization theorem fails. In section 5, we introduce new graphical criteria that preclude recoverability of joint and conditional distributions. In section 6, we discuss recoverability of causal queries and show that unlike probabilistic queries, $P(y|do(x))$ may be recovered even when $Y$ and its missingness mechanism $(R_y)$ are not d-separable. In section 7, we demonstrate how we can apply our results to problems of attrition in which missingness is a severe obstacle to sound inferences. Related works are discussed in section 8 and conclusions are drawn in section 9. Proofs of all theoretical results in this paper are provided in the appendix.

## 2 Missingness Graph and Recoverability

Missingness graphs as discussed below was first defined in Mohan et al. [2013] and we adopt the same notations. Let $G(\mathbb{V}, E)$ be the causal DAG where $\mathbb{V} = V \cup U \cup V^* \cup \mathbb{R}$. $V$ is the set of observable nodes. Nodes in the graph correspond to variables in the data set. $U$ is the set of unobserved nodes (also called latent variables). $E$ is the set of edges in the DAG. We use bi-directed edges as a shorthand notation to denote the existence of a $U$ variable as common parent of two variables in $V \cup \mathbb{R}$. $V$ is partitioned into $V_o$ and $V_m$ such that $V_o \subseteq V$ is the set of variables that are observed in all records in the population and $V_m \subseteq V$ is the set of variables that are missing in at least one record. Variable $X$ is termed as *fully observed* if $X \in V_o$, *partially observed* if $X \in V_m$ and *substantive* if $X \in V_o \cup V_m$. Associated with every partially observed variable $V_i \in V_m$ are two other variables $R_{v_i}$ and $V_i^*$, where $V_i^*$ is a proxy variable that is actually observed, and $R_{v_i}$ represents the status of the causal mechanism responsible for the missingness of $V_i^*$; formally,

$$v_i^* = f(r_{v_i}, v_i) = \begin{cases} v_i & \text{if } r_{v_i} = 0 \\ m & \text{if } r_{v_i} = 1 \end{cases} \tag{1}$$

$V^*$ is the set of all proxy variables and $\mathbb{R}$ is the set of all causal mechanisms that are responsible for missingness. $R$ variables may not be parents of variables in $V \cup U$. We call this graphical representation **Missingness Graph** (or $m$-graph). An example of an m-graph is given in Figure 1 (a).We use the following shorthand. For any variable $X$, let $X'$ be a shorthand for $X = 0$. For any set $W \subseteq V_m \cup V_o \cup R$, let $W_r$, $W_o$ and $W_m$ be the shorthand for $W \cap R$, $W \cap V_o$ and $W \cap V_m$ respectively. Let $R_w$ be a shorthand for $R_{V_m \cap W}$ i.e. $R_w$ is the set containing missingness mechanisms of all partially observed variables in $W$. Note that $R_w$ and $W_r$ are not the same. $G_{\underline{X}}$ and $G_{\overline{X}}$ represent graphs formed by removing from $G$ all edges leaving and entering $X$, respectively.

A *manifest distribution* $P(V_o, V^*, R)$ is the distribution that governs the available dataset. An *underlying distribution* $P(V_o, V_m, R)$ is said to be compatible with a given manifest distribution $P(V_o, V^*, R)$ if the latter can be obtained from the former using equation 1. Manifest distribution $P_m$ is compatible with a given underlying distribution $P_u$ if $\forall X, X \subseteq$

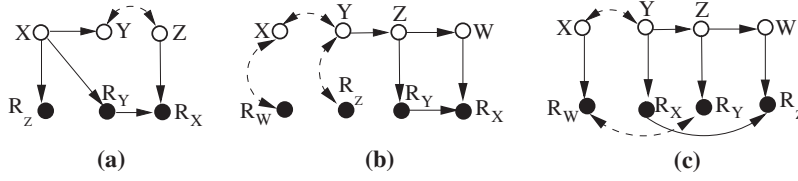

Figure 2: (a) m-graph in which $P(V)$ is recoverable by the sequential factorization (b) & (c): m-graphs for which no admissible sequence exists.

$V_m$ and $Y = V_m \setminus X$, the following equality holds true.

$$P_m(R'_x, R_y, X^*, Y^*, V_o) = P_u(R'_x, R_y, X, V_o)$$

where $R'_x$ denotes $R_x = 0$ and $R_y$ denotes $R_y = 1$. Refer Appendix 10.2 for an example.

## 2.1 Recoverability

Given a manifest distribution $P(V^*, V_o, R)$ and an m-graph $G$ that depicts the missingness process, query $Q$ is recoverable if we can compute a consistent estimate of $Q$ as if no data were missing. Formally,

**Definition 1** (Recoverability (Mohan et al. [2013])). *Given a m-graph $G$, and a target relation $Q$ defined on the variables in $V$, $Q$ is said to be recoverable in $G$ if there exists an algorithm that produces a consistent estimate of $Q$ for every dataset $D$ such that $P(D)$ is (1) compatible with $G$ and (2) strictly positive[1] over complete cases i.e. $P(V_o, V_m, \mathbb{R} = 0) > 0$.*

For an introduction to the notion of recoverability see, Pearl and Mohan [2013] and Mohan et al. [2013].

## 3 Recovering Probabilistic Queries by Sequential Factorization

Mohan et al. [2013] (theorem-4) presented a sufficient condition for recovering probabilistic queries such as joint and conditional distributions by using ordered factorizations. However, the theorem is not applicable to certain classes of problems such as those in longitudinal studies in which edges exist between $R$ variables. General ordered factorization defined below broadens the concept of ordered factorization (Mohan et al. [2013]) to include the set of $R$ variables. Subsequently, the modified theorem (stated below as theorem 1) will permit us to handle cases in which $R$ variables are contained in separating sets that d-separate partially observed variables from their respective missingness mechanisms (example: $X \perp\!\!\!\perp R_x | R_y$ in figure 2 (a)).

**Definition 2** (General Ordered factorization). *Given a graph $G$ and a set $O$ of ordered $V \cup R$ variables $Y_1 < Y_2 < \ldots < Y_k$, a general ordered factorization relative to $G$, denoted by $f(O)$, is a product of conditional probabilities $f(O) = \prod_i P(Y_i|X_i)$ where $X_i \subseteq \{Y_{i+1}, \ldots, Y_n\}$ is a minimal set such that $Y_i \perp\!\!\!\perp (\{Y_{i+1}, \ldots, Y_n\} \setminus X_i)|X_i$ holds in $G$.*

**Theorem 1** (Sequential Factorization ). *A sufficient condition for recoverability of a relation $Q$ defined over substantive variables is that $Q$ be decomposable into a general ordered factorization, or a sum of such factorizations, such that every factor $Q_i = P(Y_i|X_i)$ satisfies, (1) $Y_i \perp\!\!\!\perp (R_{y_i}, R_{x_i})|X_i \setminus \{R_{y_i}, R_{x_i}\}$, if $Y_i \in (V_o \cup V_m)$ and (2) $Z \notin X_i$ and $X_r \cap R_{X_m} = \emptyset$ and $R_z \perp\!\!\!\perp R_{X_i}|X_i$ if $Y_i = R_z$ for any $Z \in V_m$.*

An ordered factorization that satisfies the condition in Theorem 1 is called an *admissible sequence*.

The following example illustrates the use of theorem 1 for recovering the joint distribution. Additionally, it sheds light on the need for the notion of *minimality* in definition 2.

**Example 1.** *We are interested in recovering $P(X, Y, Z)$ given the m-graph in Figure 2 (a). We discern from the graph that definition 2 is satisfied because: (1) $P(Y|X, Z, R_y) = P(Y|X, Z)$ and $(X, Z)$ is a minimal set such that $Y \perp\!\!\!\perp (\{X, Z, R_y\} \setminus (X, Z))|(X, Z)$, (2) $P(X|R_y, Z) = P(X|R_y)$ and $R_y$ is the minimal set such that $X \perp\!\!\!\perp (\{R_y, Z\} \setminus R_y)|R_y$ and (3) $P(Z|R_y) = P(Z)$ and $\emptyset$ is the minimal set such that $Z \perp\!\!\!\perp R_y|\emptyset$. Therefore, the order $Y < X < Z < R_y$ induces a general ordered factorization $P(X, Y, Z, R_y) = P(Y|X, Z)P(X|R_y)P(Z)P(R_y)$. We now rewrite $P(X, Y, Z)$ as follows:*

$$P(X, Y, Z) = \sum\nolimits_{R_y} P(Y, X, Z, R_y) = P(Y|X, Z)P(Z) \sum\nolimits_{R_y} P(X|R_y)P(R_y)$$

*Since $Y \perp\!\!\!\perp R_y|X, Z$, $Z \perp\!\!\!\perp R_z$, $X \perp\!\!\!\perp R_x|R_y$, by theorem 1 we have,*

$$P(X, Y, Z) = P(Y|X, Z, R'_x, R'_y, R'_z)P(Z|R'_z) \sum\nolimits_{R_y} P(X|R'_x, R_y)P(R_y)$$

*Indeed, equation 1 permits us to rewrite it as:*

$$P(X, Y, Z) = P(Y^*|X^*, Z^*, R'_x, R'_y, R'_z)P(Z^*|R'_z) \sum\nolimits_{R_y} P(X^*|R'_x, R_y)P(R_y)$$

*$P(X, Y, Z)$ is recoverable because every term in the right hand side is consistently estimable from the available dataset.*

*Had we ignored the minimality requirement in definition 2 and chosen to factorize $Y < X < Z < R_y$ using the chain rule, we would have obtained: $P(X, Y, Z, R_y) = P(Y|X, Z, R_y)P(X|Z, R_y)P(Z|R_y)P(R_y)$ which is not admissible since $X \perp\!\!\!\perp (R_z, R_x)|Z$ does not hold in the graph. In other words, existence of one admissible sequence based on an order $O$ of variables does not guarantee that every factorization based on $O$ is admissible; it is for this reason that we need to impose the condition of minimality in definition 2.*

The recovery procedure presented in example 1 requires that we introduce $R_y$ into the order. Indeed, there is no ordered factorization over the substantive variables $\{X, Y, Z\}$ that will permit recoverability of $P(X, Y, Z)$ in figure 2 (a). This extension of Mohan et al. [2013] thus permits the recovery of probabilistic queries from problems in which the missingness mechanisms interact with one another.

## 4   Recoverability in the Absence of an Admissible Sequence

Mohan et al. [2013] presented a theorem (refer appendix 10.4) that stated the necessary and sufficient condition for recovering the joint distribution for the class of problems in which the parent set of every $R$ variable is a subset of $V_o \cup V_m$. In contrast to Theorem 1, their theorem can handle problems for which no admissible sequence exists. The following theorem gives a generalization and is applicable to any given semi-markovian model (for example, m-graphs in figure 2 (b) & (c)). It relies on the notion of collider path and two new subsets, $R^{(part)}$: the partitions of $R$ variables and $Mb(R^{(i)})$: substantive variables related to $R^{(i)}$, which we will define after stating the theorem.

**Theorem 2.** *Given an m-graph $G$ in which no element in $V_m$ is either a neighbor of its missingness mechanism or connected to its missingness mechanism by a collider path, $P(V)$ is recoverable if no $Mb(R^{(i)})$ contains a partially observed variable $X$ such that $R_x \in R^{(i)}$ i.e. $\forall i$, $R^{(i)} \cap R_{Mb(R^{(i)})} = \emptyset$. Moreover, if recoverable, $P(V)$ is given by,*

$$P(V) = \frac{P(V, R = 0)}{\prod_i P(R^{(i)} = 0|Mb(R^{(i)}), R_{Mb(R^{(i)})} = 0)}$$

In theorem 2:
  (i) collider path $p$ between any two nodes $X$ and $Y$ is a path in which every intermediate node is a collider. Example, $X \rightarrow Z < --> Y$.
  (ii) $R^{part} = \{R^{(1)}, R^{(2)}, ... R^{(N)}\}$ are partitions of $R$ variables such that for every element $R_x$ and $R_y$ belonging to distinct partitions, the following conditions hold true: (i) $R_x$ and

$R_y$ are not neighbors and (ii) $R_x$ and $R_y$ are not connected by a collider path. In figure 2 (b): $R^{part} = \{R^{(1)}, R^{(2)}\}$ where $R^{(1)} = \{R_w, R_z\}$, $R^{(2)} = \{R_x, R_y\}$

(iii) $Mb(R^{(i)})$ is the markov blanket of $R^{(i)}$ comprising of all substantive variables that are either neighbors or connected to variables in $R^{(i)}$ by a collider path (Richardson [2003]). In figure 2 (b): $Mb(R^{(1)}) = \{X, Y\}$ and $Mb(R^{(2)}) = \{Z, W\}$.

Appendix 10.6 demonstrates how theorem 2 leads to the recoverability of $P(V)$ in figure 2, to which theorems in Mohan et al. [2013] do not apply.

The following corollary yields a sufficient condition for recovering the joint distribution from the class of problems in which no bi-directed edge exists between variables in sets $R$ and $V_o \cup V_m$ (for example, the m-graph described in Figure 2 (c)). These problems form a subset of the class of problems covered in theorem 2. Subset $Pa^{sub}(R^{(i)})$ used in the corollary is the set of all substantive variables that are parents of variables in $R^{(i)}$. In figure 2 (b): $Pa^{sub}(R^{(1)}) = \emptyset$ and $Pa^{sub}(R^{(2)}) = \{Z, W\}$.

**Corollary 1.** *Let $G$ be an m-graph such that (i) $\forall X \in V_m \cup V_o$, no latent variable is a common parent of $X$ and any member of $R$, and (ii) $\forall Y \in V_m$, $Y$ is not a parent of $R_y$. If $\forall i$, $Pa^{sub}(R^{(i)})$ does not contain a partially observed variables whose missing mechanism is in $R^{(i)}$ i.e. $R^{(i)} \cap R_{Pa^{sub}(R^{(i)})} = \emptyset$, then $P(V)$ is recoverable and is given by,*

$$P(v) = \frac{P(R=0,V)}{\prod_i P(R^{(i)}=0|Pa^{sub}(R^{(i)}),R_{Pa^{sub}(R^{(i)})}=0)}$$

# 5 Non-recoverability Criteria for Joint and Conditional Distributions

Up until now, we dealt with sufficient conditions for recoverability. It is important however to supplement these results with criteria for non-recoverability in order to alert the user to the fact that the available assumptions are insufficient to produce a consistent estimate of the target query. Such criteria have not been treated formally in the literature thus far. In the following theorem we introduce two graphical conditions that preclude recoverability.

**Theorem 3** (Non-recoverability of $P(V)$). *Given a semi-markovian model $G$, the following conditions are necessary for recoverability of the joint distribution:*
*(i) $\forall X \in V_m$, $X$ and $R_x$ are not neighbors and*
*(ii) $\forall X \in V_m$, there does not exist a path from $X$ to $R_x$ in which every intermediate node is both a collider and a substantive variable.*

In the following corollary, we leverage theorem 3 to yield necessary conditions for recovering conditional distributions.

**Corollary 2.** *[Non-recoverability of $P(Y|X)$] Let $X$ and $Y$ be disjoint subsets of substantive variables. $P(Y|X)$ is non-recoverable in m-graph $G$ if one of the following conditions is true:*
*(1) $Y$ and $R_y$ are neighbors*
*(2) $G$ contains a collider path $p$ connecting $Y$ and $R_y$ such that all intermediate nodes in $p$ are in $X$.*

# 6 Recovering Causal Queries

Given a causal query and a causal bayesian network a complete algorithm exists for deciding whether the query is identifiable or not (Shpitser and Pearl [2006]). Obviously, a query that is not identifiable in the substantive model is not recoverable from missing data. Therefore, a necessary condition for recoverability of a causal query is its identifiability which we will assume in the rest of our discussion.

**Definition 3** (Trivially Recoverable Query). *A causal query $Q$ is said to be trivially recoverable given an m-graph $G$ if it has an estimand (in terms of substantive variables) in which every factor is recoverable.*

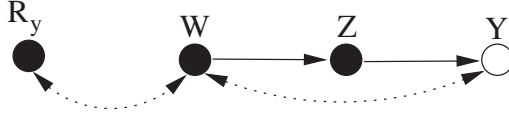

Figure 3: m-graph in which $Y$ and $R_y$ are not separable but still $P(Y|do(Z))$ is recoverable.

Classes of problems that fall into the MCAR (Missing Completely At Random) and MAR (Missing At Random) category are much discussed in the literature ((Rubin [1976])) because in such categories probabilistic queries are recoverable by graph-blind algorithms. An immediate but important implication of trivial recoverability is that if data are MAR or MCAR and the query is identifiable, then it is also recoverable by model-blind algorithms.

**Example 2.** *In the gender wage-gap study example in Figure 1 (a), the effect of sex on income, $P(I|do(S))$, is identifiable and is given by $P(I|S)$. By theorem 2, $P(S, X, Q, I)$ is recoverable. Hence $P(I|do(S))$ is recoverable.*

## 6.1   Recovering $P(y|do(z))$ when **Y** and $R_y$ are inseparable

The recoverability of $P(V)$ hinges on the separability of a partially observed variable from its missingness mechanism (a condition established in theorem 3). Remarkably, causal queries may circumvent this requirement. The following example demonstrates that $P(y|do(z))$ is recoverable even when $Y$ and $R_y$ are not separable.

**Example 3.** *Examine Figure 3. By backdoor criterion, $P(y|do(z)) = \sum_w P(y|z, w)P(w)$. One might be tempted to conclude that the causal relation is non-recoverable because $P(w, z, y)$ is non-recoverable (by theorem 2) and $P(y|z, w)$ is not recoverable (by corollary 2). However, $P(y|do(z))$ is recoverable as demonstrated below:*

$$P(y|do(z)) = P(y|do(z), R'_y) = \sum_w P(y|do(z), w, R'_y)P(w|do(z), R'_y) \qquad (2)$$

$$P(y|do(z), w, R'_y) = P(y|z, w, R'_y) \text{ (by Rule-2 of do-calculus (Pearl [2009]))} \qquad (3)$$

$$P(w|do(z), R'_y) = P(w|R'_y) \text{ (by Rule-3 of do-calculus) )} \qquad (4)$$

*Substituting (3) and (4) in (2) we get:*

$$P(y|do(z)) = \sum_w P(y|z, w, R'_y)P(w|R'_y) = \sum_w P(y^*|z, w, R'_y)P(w|R'_y)$$

The recoverability of $P(y|do(z))$ in the previous example follows from the notion of d*-separability and dormant independence [Shpitser and Pearl, 2008].

**Definition 4** ($d^*$-separation (Shpitser and Pearl [2008])). *Let $G$ be a causal diagram. Variable sets $X$, $Y$ are $d^*$-separated in $G$ given $Z$, $W$ (written $X \perp\!\!\!\perp_w Y|Z$), if we can find sets $Z, W$, such that $X \perp\!\!\!\perp Y|Z$ in $G_{\overline{w}}$, and $P(y, x|z, do(w))$ is identifiable.*

**Definition 5** (Inducing path (Verma and Pearl [1991])). *An path $p$ between $X$ and $Y$ is called inducing path if every node on the path is a collider and an ancestor of either $X$ or $Y$.*

**Theorem 4.** *Given an m-graph in which $|V_m| = 1$ and $Y$ and $R_y$ are connected by an inducing path, $P(y|do(x))$ is recoverable if there exists $Z, W$ such that $Y \perp\!\!\!\perp_w R_y|Z$ and for $W = W \setminus X$, the following conditions hold:*
*(1) $Y \perp\!\!\!\perp W_1|X, Z$ in $G_{\overline{X}, W_1}$ and*
*(2) $P(W_1, Z|do(X))$ and $\overline{P}(Y|do(W_1), do(X), Z, R'y)$ are identifiable.*
*Moreover, if recoverable then,*
$$P(y|do(x)) = \sum_{W_1, Z} P(Y|do(W), do(X), Z, R'_y)P(Z, W_1|do(X))$$

We can quickly conclude that $P(y|do(z))$ is recoverable in the m-graph in figure 3 by verifying that the conditions in theorem 4 hold in the m-graph.

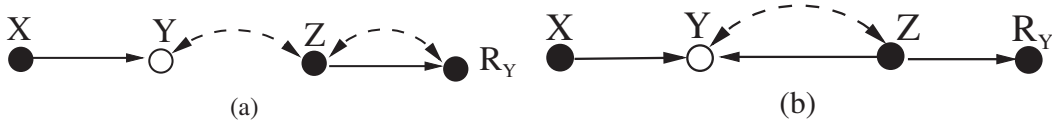

Figure 4: (a) m-graphs in which $P(y|do(x))$ is not recoverable (b) m-graphs in which $P(y|do(x))$ is recoverable.

# 7   Attrition

Attrition (i.e. participants dropping out from a study/experiment), is a ubiquitous phenomenon, especially in longitudinal studies. In this section, we shall discuss a special case of attrition called 'Simple Attrition' (Garcia [2013]). In this problem, a researcher conducts a randomized trial, measures a set of variables (X,Y,Z) and obtains a dataset where outcome (Y) is corrupted by missing values (due to attrition). Clearly, due to randomization, the effect of treatment (X) on outcome (Y), $P(y|do(x))$, is identifiable and is given by $P(Y|X)$. We shall now demonstrate the usefulness of our previous discussion in recovering $P(y|do(x))$. Typical attrition problems are depicted in figure 4. In Figure 4 (b) we can apply theorem 1 to recover $P(y|do(x))$ as given below: $P(Y|X) = \sum_Z P(Y^*|X, Z, R'_y)P(Z|X)$. In Figure 4 (a), we observe that $Y$ and $R_y$ are connected by a collider path. Therefore by corollary 2, $P(Y|X)$ is not recoverable; hence $P(y|do(x))$ is also not recoverable.

## 7.1   Recovering Joint Distributions under simple attrition

The following theorem yields the *necessary and sufficient* condition for recovering joint distributions from semi-markovian models with a single partially observed variable i.e. $|V_m| = 1$ which includes models afflicted by simple attrition.

**Theorem 5.** *Let $Y \in V_m$ and $|V_m| = 1$. $P(V)$ is recoverable in m-graph $G$ if and only if $Y$ and $R_y$ are not neighbors and $Y$ and $R_y$ are not connected by a path in which all intermediate nodes are colliders. If both conditions are satisfied, then $P(V)$ is given by, $P(V) = P(Y|V_O, R_y = 0)P(V_O)$*

## 7.2   Recovering Causal Effects under Simple Attrition

**Theorem 6.** *$P(y|do(x))$ is recoverable in the simple attrition case (with one partially observed variable) if and only if $Y$ and $R_y$ are neither neighbors nor connected by an inducing path. Moreover, if recoverable,*

$$P(Y|X) = \sum_z P(Y^*|X, Z, R'_y)P(Z|X) \qquad (5)$$

*where $Z$ is the separating set that d-separates $Y$ from $R_y$.*

These results rectify prevailing opinion in the available literature. For example, according to Garcia [2013] (Theorem-3), a necessary condition for non-recoverability of causal effect under simple attrition is that $X$ be an ancestor of $R_y$. In Figure 4 (a), $X$ is not an ancestor of $R_y$ and still $P(Y|X)$ is non-recoverable ( due to the collider path between $Y$ and $R_y$ ).

# 8   Related Work

Deletion based methods such as listwise deletion that are easy to understand as well as implement, guarantee consistent estimates only for certain categories of missingness such as MCAR (Rubin [1976]). Maximum Likelihood method is known to yield consistent estimates under MAR assumption; expectation maximization algorithm and gradient based algorithms are widely used for searching for ML estimates under incomplete data (Lauritzen [1995], Dempster et al. [1977], Darwiche [2009], Koller and Friedman [2009]). Most work in machine learning assumes MAR and proceeds with ML or Bayesian inference. However, there are exceptions such as recent work on collaborative filtering and recommender systems which

develop probabilistic models that explicitly incorporate missing data mechanism (Marlin et al. [2011], Marlin and Zemel [2009], Marlin et al. [2007]).

Other methods for handling missing data can be classified into two: (a) Inverse Probability Weighted Methods and (b) Imputation based methods (Rothman et al. [2008]). Inverse Probability Weighing methods analyze and assign weights to complete records based on estimated probabilities of completeness (Van der Laan and Robins [2003], Robins et al. [1994]). Imputation based methods substitute a reasonable guess in the place of a missing value (Allison [2002]) and Multiple Imputation (Little and Rubin [2002]) is a widely used imputation method.

Missing data is a special case of coarsened data and data are said to be coarsened at random (CAR) if the coarsening mechanism is only a function of the observed data (Heitjan and Rubin [1991]). Robins and Rotnitzky [1992] introduced a methodology for parameter estimation from data structures for which full data has a non-zero probability of being fully observed and their methodology was later extended to deal with censored data in which complete data on subjects are never observed (Van Der Laan and Robins [1998]).

The use of graphical models for handling missing data is a relatively new development. Daniel et al. [2012] used graphical models for analyzing missing information in the form of missing cases (due to sample selection bias). Attrition is a common occurrence in longitudinal studies and arises when subjects drop out of the study (Twisk and de Vente [2002], Shadish [2002]) and Garcia [2013] analysed the problem of attrition using causal graphs. Thoemmes and Rose [2013] cautioned the practitioner that contrary to popular belief, not all auxiliary variables reduce bias. Both Garcia [2013] and Thoemmes and Rose [2013] associate missingness with a single variable and interactions among several missingness mechanisms are unexplored.

Mohan et al. [2013] employed a formal representation called Missingness Graphs to depict the missingness process, defined the notion of recoverability and derived conditions under which queries would be recoverable when datasets are categorized as Missing Not At Random (MNAR). Tests to detect misspecifications in the m-graph are discussed in Mohan and Pearl [2014].

## 9    Conclusion

Graphical models play a critical role in portraying the missingness process, encoding and communicating assumptions about missingness and deciding recoverability given a dataset afflicted with missingness. We presented graphical conditions for recovering joint and conditional distributions and sufficient conditions for recovering causal queries. We exemplified the recoverability of causal queries of the form $P(y|do(x))$ despite the existence of an inseparable path between $Y$ and $R_y$, which is an insurmountable obstacle to the recovery of P(Y). We applied our results to problems of attrition and presented necessary and sufficient graphical conditions for recovering causal effects in such problems.

## Acknowledgement

This paper has benefited from discussions with Ilya Shpitser. This research was supported in parts by grants from NSF #IIS1249822 and #IIS1302448, and ONR #N00014-13-1-0153 and #N00014-10-1-0933.

## Footnotes

[1]An extension to datasets that are not strictly positive over complete cases is sometimes feasible(Mohan et al. [2013]).

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
