[Supplementary Material]

# 10    Appendix

## 10.1    Missingness Process in Figure 1

Figure 1 Missingness Graph depicting the missingness process in a hypothetical (job-specific) gender wage gap study that measured the variables: sex (S), work experience(X), qualification(Q) and income(I). Fully observed and partially observed variables are represented by filled and hollow nodes respectively. While sex and work experience were found to be fully observed in all records i.e. $V_o = \{S, X\}$, qualification and income were found to be missing in some of the records i.e. $V_m = \{Q, I\}$. $R_Q$ and $R_I$ denote the causes of missingness of Q and I respectively and are assumed to be independent of S,Q,I and X. The assumptions in the model are: (1) women are likely to be less qualified and experienced than men, (2) income is determined by qualification and job experience of the candidate, and (3) missingness in Q and I are correlated, caused by unobserved common factors such as laziness or resistance to respond.

## 10.2    Testing compatibility between underlying and manifest distributions

**Example 4.** *Let the incomplete dataset contain two partially observed variables, Z and W. The tests for compatibility between manifest distribution: $P_m(Z^*, W^*, R_z, R_w)$ and the underlying distribution: $P_u(Z, W, R_z, R_w)$ are:*

**Case-1:** *Let $X = \{Z, W\}$, then $Y = V_m \setminus X = \{\}$*
$P_m(Z^* = z, W^* = w, R_z = 0, R_w = 0) = P_u(Z = z, W = w, R_z = 0, R_w = 0) \forall z, w$

**Case-2:** *Let $X = \{Z\}$, then $Y = \{W\}$*
$P_m(Z^* = z, W^* = m, R_z = 0, R_w = 1) = \sum_w P_u(Z = z, w, R_z = 0, R_w = 1) \forall z$

**Case-3:** *Let $X = \{W\}$, then $Y = \{Z\}$*
$P_m(Z^* = m, W^* = w, R_z = 1, R_w = 0) = \sum_z P_u(z, W = w, R_z = 1, R_w = 0) \forall w$

**Case-4:** *Let $X = \{\}$, then $Y = \{Z, W\}$*
$P_m(Z^* = m, W^* = m, R_z = 1, R_w = 1) = \sum_{z,w} P_u(z, w, R_z = 1, R_w = 1)$

## 10.3    Proof of theorem 1

*Proof.* follows from Theorem-1 in Mohan et al. [2013] (restated below as theorem 7) noting that ordered factorization is one specific form of decomposition.    □

**Theorem 7** (Mohan et al. [2013]). *A query Q defined over variables in $V_o \cup V_m$ is recoverable if it is decomposable into terms of the form $Q_j = P(S_j | T_j)$ such that $T_j$ contains the missingness mechanism $R_v = 0$ of every partially observed variable V that appears in $Q_j$.*

## 10.4    Recovering $P(V)$ when parents of $R$ belong to $V_o \cup V_m$

**Theorem 8** (Recoverability of the Joint $P(V)$ (Mohan et al. [2013])). *Given a m-graph G with no edges between the R variables and no latent variables as parents of R variables, a necessary and sufficient condition for recovering the joint distribution $P(V)$ is that no variable X be a parent of its missingness mechanism $R_X$. Moreover, when recoverable, $P(V)$ is given by*

$$P(v) = \frac{P(R = 0, v)}{\prod_i P(R_i = 0 | pa^o_{r_i}, pa^m_{r_i}, R_{Pa^m_{r_i}} = 0)} \tag{6}$$

*where $Pa^o_{r_i} \subseteq V_o$ and $Pa^m_{r_i} \subseteq V_m$ are the parents of $R_i$.*

**Example 5.** *We wish to recover $P(X, Y, Z)$ from the m-graph in Figure 1 (a). An enumeration of various orderings will reveal that none of the orders are admissible. Nevertheless, using theorem 8, we can recover the joint probability as given below:*

$$P(X, Y, Z) = \frac{P(R'_x, R'_y, R'_z, X, Y, Z)}{P(R'_z | X, R'_x) P(R'_x | Y, R'_y) P(R'_y | Z, R'_z)}$$

Figure 5: m-graph in which joint distribution is recoverable.

## 10.5 Proof of Theorem 2

*Proof.*

$$P(V) = \frac{P(R=0,V)}{P(R=0|V)}$$

$$= \frac{P(R=0,V)}{P(R^{(1)}=0, R^{(2)}=0, ...R^N=0|V)}$$

$Mb(R^{(i)})$ d-separates $R^{(i)}$ from all variables that are not in $R^{(i)} \cup Mb(R^{(i)})$ i.e. $R^{(i)} \perp\!\!\!\perp (\{R, V\} - \{R^{(i)}, Mb(R^{(i)})\})|Mb(R^{(i)})$ . Hence,

$$P(V) = \frac{P(R=0,V)}{\prod_i P(R^{(i)}=0|Mb(R^{(i)}))}$$

Using $R^{(i)} \cap R_{Mb(R^{(i)})} = \emptyset$ and $R^{(i)} \perp\!\!\!\perp (\{R,V\} - \{R^{(i)}, Mb(R^{(i)})\})|Mb(R^{(i)})$ we get,

$$P(V) = \frac{P(R=0,V)}{\prod_i P(R^{(i)}=0|Mb(R^{(i)}), R_{Mb(R^{(i)})}=0)}$$

Now we can directly apply equation 1 and express $P(V)$ in terms of quantities estimable from the available dataset. Therefore, $P(V)$ is recoverable. $\square$

## 10.6 Example: Recoverability by Theorem 2

**Example 6.** $P(X, Y, Z, W)$ *is the query of interest and Figure 2 (b) depicts the missingness process and identifies the sets $R^{part}$ and $Mb(R^{(i)})$. A quick inspection reveals that no admissible sequence exists. However, notice that $CI_1$ : $R^{(1)} \perp\!\!\!\perp (R^{(2)}, Mb(R^{(2)}))|Mb(R^{(1)})$ and $CI_2$ : $R^{(2)} \perp\!\!\!\perp (R^{(1)}, Mb(R^{(1)}))|Mb(R^{(2)})$ hold in the m-graph. We exploit these independencies to recover the joint distribution as detailed below:*

$$P(X,Y,Z,W) = \frac{P(R=0,X,Y,Z,W)}{P(R=0|X,Y,Z,W)} = \frac{P(R=0,X,Y,Z,W)}{P(R^{(1)}=0,R^{(2)}=0|X,Y,Z,W)}$$

$$= \frac{P(R=0,X,Y,Z,W)}{P(R^{(1)}=0|X,Y,R^{(2)}=0)P(R^{(2)}=0|Z,W,R^{(1)}=0)} \; (Using\ CI_1\ and\ CI_2)$$

$$P(V) = \frac{P(R=0,X^*,Y^*,Z^*,W^*)}{P(R_w=0,R_z=0|X^*,Y^*,R_x=0,R_y=0)P(R_x=0,R_y=0|Z^*,W^*,R_z=0,R_w=0)} \; (By\ equation\ 1)$$

## 10.7 Proof of Corollary 1

*Proof.*

$$P(V) = \frac{P(R=0,V)}{P(R=0|V)}$$

$$= \frac{P(R=0,V)}{P(R^{(1)}, R^{(2)}, ...R^N|V)}$$

Since $Pa^{sub}(R^{(i)}) \subseteq V$ d-separates $R_i$ from all the other variables in $(V \cup R) \setminus (R^{(i)} \cup Pa^{sub}(R^{(i)}))$ , we get

$$P(V) = \frac{P(R = 0, V)}{\prod_i P(R^{(i)} = 0 | Pa^{sub}(R^{(i)}))}$$

Using $R^{(i)} \cap R_{Pa^{sub}(R^{(i)})} = \emptyset$ and $R^{(i)} \perp\!\!\!\perp (\{R, V\} - \{R^{(i)}, Pa^{sub}(R^{(i)})\}) | Pa^{sub}(R^{(i)})$ we get,

$$P(V) = \frac{P(R = 0, V)}{\prod_i P(R^{(i)} = 0 | Pa^{sub}(R^{(i)}), R_{Pa^{sub}(R^{(i)})} = 0)}$$

$\square$

## 10.8   Proof of Theorem 3

We will be using the following lemma (stated and proved in Mohan et al. [2013] (Supplementary materials)) in our proof.

**Lemma 1.** *If a target relation $Q$ is not recoverable in m-graph $G$, then $Q$ is not recoverable in the graph $G'$ resulting from adding a single edge to $G$.*

*Proof.* Non-recoverability of $P(V)$ when $X$ is a parent of $R_x$ has been proved in Mohan et al. [2013]. If $P(V)$ is non-recoverable when $G$ contains subgraph $G_1 : X \to R_x$, then $P(V)$ is non-recoverable when $G$ contains subgraph $G_2 : X < --U-- > R_x$ since, (a) $G_1$ and $G_2$ are equivalent models and (b) we are dealing with recoverability of a probabilistic query. Nevertheless, a detailed proof by construction follows.

$M_1$ and $M_2$ are two models in which variables $U, X$ and $R_x$ are binary and $U$ is a fair coin. In $M_1$, $X = 0$ and $R_x = u$ and in $M_2$, $X = u$ and $R_x = u$. Notice that although the two models agree on the manifest distribution, they disagree on the query $P(X)$. Hence $P(X)$ is non-recoverable in $X < --U-- > R_x$. Using Lemma-1 (Refer appendix), we can conclude that $P(V)$ is non-recoverable in any m-graph in which $X$ and $R_x$ are connected by a bi-directed edge.

Figure 6: An m-graph in which $P(X, Z)$ is not-recoverable where $Z = \{Z_1, Z_2, ..., Z_k\}$. $X$ is partially observed, all $Z$ variables are fully observed, parents of $Z_i$ are $U_{i-1}$ and $U_i$, parent of $X$ is $U_o$ and parent of $R_x$ is $U_k$.

Given the m-graph in Figure 6 we will now prove that $P(X, Z_1, Z_2...Z_k)$ is non-recoverable. Let $M_3$ and $M_4$ be two models such that all the variables are binary, all the U variables are fair coins, $X = U_0$, $R_x = U_k$ and $Z_i = U_{i-1} \oplus U_i$, $1 \le i < k$. In $M_3$, $Z_k = U_{k-1}$ and in $M_4$, $Z_k = U_{k-1} \oplus U_k$. Both models yield the same manifest distribution. However, they disagree on the query $P(X, Z_1, Z_2...Z_k)$. For instance, in $M_3$, $P(X = 0, Z = 0, R_x = 1) > 0$ where as in $M_4$, $P(X = 0, Z = 0, R_x = 1) = 0$. Therefore in $M_4$, $P(X = 0, Z = 0) = P(X = 0, Z = 0, R_x = 0)$ and in $M_3$, $P(X = 0, Z = 0) = P(X = 0, Z = 0, R_x = 0) + P(X = 0, Z = 0, R_x = 1)$. Hence in the m-graph in figure 6, the joint distribution $P(X, Z)$ is non-recoverable. Using lemma 1, we can conclude that joint distribution is non-recoverable in any m-graph which has a bi-directed path from any partially observed variable $X$ to its missingness mechanism $R_x$. $\square$

## 10.9   Proof of Corollary 2

*Proof.* Let $|V_m| = 1$ and $Y_1 \in Y$ be the only partially observed variable. Let $G'$ be the subgraph containing all variables in $X \cup Y \cup \{R_{y_1}, Y_1^*\}$. We know that if (1) or (2) are true,

then, (i) $P(X, Y)$ is not recoverable in $G'$ and (ii) $P(X)$ is recoverable in $G'$. Therefore, $P(Y|X) = \frac{P(Y,X)}{P(X)}$ is not recoverable in $G'$ and hence by lemma 1, not recoverable in $G$. $\square$

### 10.10  Proof of Theorem 4

*Proof.* $P(Y|do(X)) = \sum_{z,w'} P(Y|Z, W', do(X))P(Z, W'|do(X))$
If condition 1 holds, then by Rule-2 of do-calculus (Pearl [2009]) we have:
$P(Y|Z, W', do(X)) = P(Y|Z, do(X), do(W'))$
Since $Y \perp\!\!\!\perp_w R_y|Z$,
$$P(Y|Z, do(X), do(W')) = P(Y|Z, do(X), do(W'), R_y')$$
$$= P(Y^*|Z, do(X), do(W'), R_y')$$
Therefore, $P(y|do(x))$ is recoverable. $\square$

### 10.11  Proof of Theorem 5

*Proof.* (sufficiency) Whenever (1) and (2) are satisfied, $Y \perp\!\!\!\perp R_y|V_o$ holds. Hence, $P(V)$ which may be written as $P(Y|V_O)P(V_O)$ can be recovered as $P(Y^*|V_O, R_y = 0)P(V_O)$.
(necessity) follows from theorem 2. $\square$

### 10.12  Proof of Theorem 6

*Proof.* (sufficiency) Under simple attrition, all paths to $R_y$ from $Y$ containing $X$ are blocked by $X$. Therefore, when both conditions specified in the theorem are satisfied, it implies that $Y$ and $R_y$ are separable. Given that $Z$ is any separator between $Y$ and $R_y$, $P(Y|X)$ may be recovered as $\sum_z P(Y^*|X, Z, R_y')P(Z|X)$.

(necessity) follows from theorem 2 $\square$