[Reviews · NeurIPS 2014]

Submitted by Assigned_Reviewer_11

The paper deals with the problem of identifiability in the presence of various generated mechanisms for missing data, using a framework introduced by Mohan and others. Some existing results are improved upon, and some nice examples are given (such as identifiability of causal distributions even when conditional distributions are unavailable).

Quality

Technical quality seems high, the results are very involved but appear to be correct. Results like Theorem 3 are clear and interpretable. Some of the results are incremental improvements upon existing ones, and don't contribute (so far as I can see) to a bigger picture of what is really going on here.

Section 3 seems orthogonal to the rest of the paper, and its main result (Theorem 1) is quite hard to comprehend. It seems to be closely related to the Markov blanket results in Richardson (2009). The quality improves later on, with section 5 giving some nice clear answers, and Section 6 some interesting results about causal queries.

---

Clarity

This is the paper's main weakness. Firstly, there is probably too much material for a conference paper here (though the importance of some of it is unclear), as evidenced by the density of the text and the 13 page technical appendix. The article does not flow clearly, and is hard to follow as a result - probably this would work better as a longer, more considered, journal paper.

Whilst finding good notation for so many quantities is not easy, the notation could certainly be improved. The use of the same capital 'V' to denote so many different objects makes it hard work for the reader, and in places the subscripts become illegible. In addition, the fact that V_i is a vertex but V_o a set of vertices is particularly confusing. The set equation in Corollary 1 is perhaps the most extreme example of poor presentation.

---

Originality and Significance

Sections 5 and 6 are clear and interesting, and potentially quite significant. This paper deals with an important problem, and contains results which seem genuinely interesting and useful.

It would perhaps be better to step back and work out what is going on in the general case, and (try to) come up with a complete criterion for identifiability in this framework (as was achieved in the case of finding causal effects without missingness). It is difficult to imagine a statistician trawling through a series of papers like this to find the precise result which shows (non)-identifiability of the causal effect she is interested in, especially if that result is as complicated as Theorem 1.

Theorem 2 and Corollary 1 seem to be a form of inverse probability weighting. The reason that these results work is presumably something to do with the conditioning sets being Markov blankets for R(i) in the mixed graphs the author is using. Richardson (2003) and (2009) might be relevant.

---

Other Comments

p4, l214: R^part _is_ a partition of R, not "a set containing partitions". The condition on line 214 should then be that Rx and Ry "belong to distinct elements of the partition".

Language is sometimes a bit casual: e.g. 'oftentimes'

p4 l213. Use \leftrightarrow rather than <-->

Is Garcia (2013) actually wrong, or just a different framework?

A notation for graphs constructed by severing edges is used in Definition 4 and Theorem 4, but not defined anywhere.

V_o and V_m are used in the sentence before they are defined (apparently unnecessarily).

---

References

Richardson, T.S., Markov properties for acyclic directed mixed graphs, SJS 30(1) 145-157, 2003

Richardson, T.S., A factorization criterion for acyclic directed mixed graphs, UAI 25, 2009.
Summary: Important problem and useful approach, though marred by poor presentation and notation.

Submitted by Assigned_Reviewer_30

This paper proposes novel necessary and sufficient conditions to guarantee whether a causal query can be inferred from data in the presence of missing observations. It extends previous works that explicitly model the missing data generating mechanism into the causal diagram. The paper is clearly written and while it builds up on previous contributions using this approach, the additional insights provided constitute, in the opinion of this reviewer, a valuable contribution to the state of the art in the analysis of causal mechanisms in the presence of missing data.
Summary: Focused paper, clearly written.

Submitted by Assigned_Reviewer_35

This is an interesting paper. It addresses the problem of when a distribution may be recovered from knowledge of the generating process and the process leading to missing data.

The paper considers a more general class of missingness process than considered in earlier work by Mohan et al.
This leads to a novel identification formula.

In addition the authors consider the identification of causal queries and show that these can be identified even when the (observational) joint distribution is not identified.

The paper is generally well-written, though there are some glitches and areas for improvement I describe below.

MAIN COMMENTS:

(1) The results here require knowledge of the process by which observations are censored.
Is this knowledge available in typical machine learning applications?
It would be helpful to include more of the detail regarding Figure 1 in the main paper. (This could be done by making the Figure smaller.)

(2) Section 2. It should be mentioned that the missing data processes here always assume that the full population is known:
Even if V_o = emptyset, and R_v =1 for all V in V_m (so no variables are observed for this individual) it is still supposed that we know this individual exists.
(This excludes missing data problems such as estimating the number of animals in a population.)

(3) The term mgraph seems to be used ambiguously. In particular, sometimes the graph includes the V* variables (Figure 1)
at other times it does not (Figure 2). If the V* variables are simply being omitted in these graphs then this should be explained.

(4) In this paper mgraphs are not allowed to have R variables as parents of variables in V or U.
Given that the authors describe treat the R variables as real quantities not simply indicators, they should explain why this restriction is justified (or whether it is mainly for convenience).

(5) The term 'neighbor', which appears at the bottom of p.4 is not defined.

(6) p. 8 first paragraph. "rectification" of the observation by Garcia.
recoverability as defined by the authors assumes positivity. Do Garcia et al also assume positivity?
If not, they might require a stronger condition.

(7) p.8 second paragraph. "Maximum Likelihood method is known to yield consistent estimates under MAR assumption".
This is much too vague. Why could maximum likelihood not be applied to any of the missingness processes described here.

MINOR COMMENTS

P.1 line 33. can bias outcomes

p.2 figure 1. Qualif*i*cations

p.2 Example 1 on p.3 line 158. P(Y |X,Z,R_y) = P(X,Z)
the RHS here should be P(Y|X,Z).

p.3 line 168. Since Y _||_ R | X,Z
this should be: Y _||_ R_y | X,Z

p.3 line 214. Condition (ii) R^{part} ={R^{(1)},R^{(2)}, \ldots , R^N}
Should be R^{(N)}.

p.3 line 215 R^j respectively
should be R^{(j)}

p.4 Corollary 1. does not contain a partially observed variables

p.6 Footnote 2. Please give a reference for the variable-level definition
Summary: An interesting paper that builds on earlier work.
Not clear how broad the application is, but technically sound.
Author Feedback
Author rebuttal: We thank the reviewers for taking time to review our paper and provide insightful comments.

Reviewer_11:

Mohan et al do not consider recoverability from the class of problems in which the missingness mechanisms (R variables) interact with each other. Given that such interactions are very common in real world applications such as attrition (and other longitudinal studies), it is imperative that we establish conditions under which recoverability is feasible when such interactions occur. Therefore, sections 3 and 4, which establish recoverability for this class of problem should not be treated as incremental improvements over existing ones. Indeed, finding sufficient conditions for systems with interacting R variables is a non-trivial challenge.

Although theorem-1 may seem daunting at first, its power should not be underestimated. In a follow up work we have capitalized on theorem-1 to develop algorithms that, (1) run in polynomial time and yield estimates for a large subset of MNAR problems in which R variables interact and (2) given sufficient data, yield consistent estimates that are orders of magnitude more accurate than EM and run orders of magnitude faster than EM for datasets falling under MAR and MCAR categories.

The reviewer is correct in saying that theorem-1 was not used for proving corollary-2. In fact there is a typo in line-256 and it is theorem-3 that is leveraged for proving corollary-2. We have made this correction.

We are considering ways of improving the notations. For example, we shall be distinguishing between sets and singleton variables using bold face.

We will include a brief discussion on the relation to Markov blankets, along the lines of Richardson's papers.

We believe that results developed thus far are non-trivial and quite useful, despite their formal incompleteness. For example, recoverability under inseparability conditions (sec:6.1) is rather surprising; it has not been reported before, and we feel they ought to be brought to attention of the research community.

Inverse Probability Weighting is mentioned in the section "related work" and we agree with the reviewer that we should make this perspective explicit in section-4 as well.

Garcia's paper mainly deals with attrition and recovery of events when missingness is negligible.

We shall define the notations for graphs constructed by severing edges that are used in definition-4 and theorem-4.

Reviewer_35:
(1) Knowledge of the process is only occassionaly available.
However even when unavailable, hypothesizing a graph structure provides significant insights since it tells the user what to watch for and what to hope for. In addition, in Mohan and Pearl (AISTATS 2014), testability results were established which allow the user to reject hypothesized models when incompatible with data. Yes, we will move the details regarding Fig-1 from the appendix to the main paper.

(2) It seems to us that the reviewer is referring to selection bias, where missingness means complete exclusion of a sample from the data set. Selection bias is handled by other methods, as in for example, Bareinboim, Tian and Pearl (AAAI 2014).

(3) Explict depiction of proxy variables are omitted for simplicity and clarity of the image. Additionally since proxy variables are always colliders, the paths traversing them are blocked.

(4) The R variables can be thought of as the last eventuality that triggers missingness and, as such, it is rarely needed for explaining other processes, if at all. Consider a person who chooses to not answer a question on a questionnaire because he is tired. Here, R is the decision to not answer the question, and tiredness is the parent of R. Tiredness can definitely trigger other processes but it is very rare that R --- the decision to not to answer this particular question -- would trigger other processes. Moreover, we do not lose any information if we attribute the trigger of those other processes to R's parent, namely "tiredness".

(6) It is hard to tell because Garcia does not take the next step of providing an estimand and this is where positivity comes into play.

(7) Our paper deals with the MNAR (Missing Not At Random) category of missingness. To the best of our knowledge there does not exist an ML based algorithm for MNAR-type problems that produces a consistent estimate whenever such exists, or even decides if a given query is recoverable or not.